# Cytotoxic activities of CD8+ T cells collaborate with macrophages to protect against blood-stage murine malaria

Takashi Imai[1], Hidekazu Ishida[2], Kazutomo Suzue[1], Tomoyo Taniguchi[1,3], Hiroko Okada[1], Chikako Shimokawa[4], Hajime Hisaeda[1]*

[1]Department of Parasitology, Gunma University Graduate School of Medicine, Maebashi, Japan; [2]Microbiological Research Institute, Otsuka Pharmaceutical Co., Ltd, Tokushima, Japan; [3]Center for Medical Education, Gunma University Faculty of Medicine, Maebashi, Japan; [4]Laboratory for Intestinal Ecosystem, RIKEN Center for Integrative Medical Science, Yokohama, Japan

**Abstract** The protective immunity afforded by CD8+ T cells against blood-stage malaria remains controversial because no MHC class I molecules are displayed on parasite-infected human erythrocytes. We recently reported that rodent malaria parasites infect erythroblasts that express major histocompatibility complex (MHC) class I antigens, which are recognized by CD8+ T cells. In this study, we demonstrate that the cytotoxic activity of CD8+ T cells contributes to the protection of mice against blood-stage malaria in a Fas ligand (FasL)-dependent manner. Erythroblasts infected with malarial parasites express the death receptor Fas. CD8+ T cells induce the externalization of phosphatidylserine (PS) on the infected erythroblasts in a cell-to-cell contact-dependent manner. PS enhances the engulfment of the infected erythroid cells by phagocytes. As a PS receptor, T-cell immunoglobulin-domain and mucin-domain-containing molecule 4 (Tim-4) contributes to the phagocytosis of malaria-parasite-infected cells. Our findings provide insight into the molecular mechanisms underlying the protective immunity exerted by CD8+ T cells in collaboration with phagocytes.

*For correspondence: hisa@gunma-u.ac.jp

## Introduction

Malaria is one of the world's three major infectious diseases, together with AIDS and tuberculosis, accounting for approximately 200 million cases annually, with 600,000 deaths (*Snow et al., 2005*; *Murray et al., 2012*). With the spread of drug-resistant parasites and the lack of effective vaccines, malaria is a serious global health problem, especially in developing countries. To develop malarial vaccines, it is necessary to understand the protective immune response against malaria. However, because the malaria parasite successfully evades the host immune responses (*Hisaeda et al., 2004*), it is difficult to identify the truly important immune responses, hindering the development of a malarial vaccine (*Good and Engwerda, 2011*).

Antibodies play a major role in the protective immunity directed against the blood-stage malaria parasite. CD4+ T cells contribute to protection against blood-stage malaria though induction of antibody production and macrophage activation (*Good and Doolan, 1999*; *Marsh and Kinyanjui, 2006*; *Jafarshad et al., 2007*; *Langhorne et al., 2008*). However, the contribution of CD8+ T cells to this protection remains controversial because there are no major histocompatibility complex (MHC) class I antigens on human erythrocytes infected with the malaria parasite. Some studies have shown that infection of BALB/c mice with non-lethal *Plasmodium yoelii* was controlled even after depletion of CD8+ T cells comparable to control mice (*Vinetz et al., 1990*). Moreover, MHC class I null mice (beta 2-microglobulin-deficient mice) recovered from infection with *Plasmodium chabaudi chabaudi* AS or *Plasmodium chabaudi adami* (*van der Heyde et al., 1993b*). Other studies have reported that

**eLife digest** The immune system consists of several different types of cell that work together to prevent infection and disease. For example, immune cells called cytotoxic CD8[+] T cells kill tumor cells or other cells that are infected. To do so, the CD8[+] T cells must recognize certain molecules on the surface of the tumor or infected cells and bind to them.

Malaria is an infectious disease caused by the *Plasmodium* parasite, which is transferred between individuals by mosquitoes. The parasite is able to evade the immune system—so much so that it is not well understood how the immune system tries to respond to stop the infection. This has made it difficult to develop a vaccine that protects against malaria.

During the latter stages of a malaria infection, the parasite infects the host's red blood cells. It was long believed that CD8[+] T cells did not help to eliminate the red blood cells that had been infected by *Plasmodium*. However, recent work in mice suggested that CD8[+] T cells do respond to infected erythroblasts—precursor cells that develop into red blood cells—and that CD8[+] T cells help protect mice against blood-stage malaria.

Now, Imai et al. describe how the CD8[+] T cells in mice help to kill erythroblasts infected with *Plasmodium yoelli*, a species of the parasite used to study malaria in mice. The infected cells display a protein called Fas on their surface. Imai et al. found that, during a malaria infection, the CD8[+] T cells produce a protein that can interact with Fas. This interaction causes the infected cell to move a signaling molecule to its outside surface, which encourages another type of immune cell to engulf and destroy the infected cell. This knowledge of how CD8[+] T cells fight *Plasmodium* parasites in the bloodstream could now help to develop new types of blood-stage vaccine for malaria.

depletion of CD8[+] T cells from mice infected with *P. chabaudi* attenuated their protection, confirming the importance of CD8[+] T cells (*Suss et al., 1988*; *Podoba and Stevenson, 1991*; *van der Heyde et al., 1993a*; *Horne-Debets et al., 2013*). However, these studies did not show the effector mechanism of CD8[+] T cells against blood-stage malaria protection.

We have conclusively demonstrated the protective roles of CD8[+] T cells using prime–boost live vaccination with the non-lethal rodent parasite *P. yoelii* 17XNL (PyNL) against challenge with the lethal *P. yoelii* 17XL (PyL) strain (*Imai et al., 2010*). The transfer of CD8[+] T cells from mice cured of PyNL infection into *Rag2*[−/−] or irradiated recipients, followed by two boosts with PyL, conferred protection against PyL. The major protective mechanism of CD8[+] T cells is the interferon γ (IFN-γ)-dependent activation of phagocytes, resulting in the enhanced phagocytosis of parasitized red blood cells (pRBCs). The cytotoxic activity of CD8[+] T cells also contributes to protecting the host against blood-stage malaria. However, the target cells of this cytotoxicity and how this cytotoxicity acts against blood-stage malaria are as yet unknown. Although recent reports have demonstrated that the human malaria parasites *Plasmodium falciparum* and *Plasmodium vivax* parasitize erythroblasts (*Ru et al., 2009*; *Tamez et al., 2009*), the host response and protective immunity against these parasitized erythroblasts are unclear. We have reported that PyNL parasites also infect erythroblasts that express MHC class I molecules on their surfaces and that CD8[+] T cells produce IFN-γ in response to parasitized erythroblasts in an antigen-specific manner. These results suggest that parasitized erythroblasts are the targets of CD8[+] T cells.

In this study, we investigated the effector mechanism of CD8[+] T cells against blood-stage malaria in detail. Splenic CD8[+] T cells activated during malaria express Fas ligand (FasL) and interact with Fas-expressing parasitized erythroblasts. As a result, phosphatidylserine (PS) is externalized to the outer leaflet of the cell membrane, leading to enhanced phagocytosis of the parasitized cells. Thus, CD8[+] T cells expressing FasL contribute to the immune response to blood-stage malaria by making parasitized cells susceptible to phagocytosis.

## Results

### Depletion of CD8[+] T cells attenuates protection against blood-stage PyNL infection

C57BL/6 mice infected with PyNL exhibited peak parasitemia of up to 30% and recovered from the infection. However, those depleted of CD8[+] T cells showed significantly greater parasitemia and died

from the infection (*Figure 1A*, *Figure 1—figure supplement 1C*). BALB/c mice depleted of CD8+ T cells showed similar results (*Figure 1—figure supplement 1D*), indicating that CD8+ T cells play an essential role in protecting mice against blood-stage malaria. CD4+ T cells are known to be important in the protective immune response to blood-stage malaria (*Suss et al., 1988*; *Kumar et al., 1989*; *Podoba and Stevenson, 1991*; *Good and Doolan, 1999*), and we confirmed that CD4+-T-cell-depleted mice displayed greater parasitemia and a higher mortality rate (*Figure 1A*). However, the course of infection clearly differed in CD8+-T-cell-depleted and CD4+-T-cell-depleted mice. Although mice depleted of CD8+ T cells suffered from much greater parasitemia from the early phase to its peak, the survivors eliminated the parasites similar to the control mice, whereas the CD4+-T-cell-depleted survivors took longer to recover from infection. This suggests that CD4+ and CD8+ T cells have different effector mechanisms for parasite clearance, and that the protective immunity mediated by CD8+ T cells is important in controlling infection during the early phase, within the period of peak parasitemia. Therefore, the following analyses were conducted 7–8 days after infection, when the CD8+ T cells might be activated in response to the parasite, and 16–18 days after infection, when the parasites begin to be eliminated.

First, we evaluated whether the activation of CD8+ T cells occurs during infection with PyNL. PyNL infection increased the proportion of CD8+ T cells that expressed activation markers such as CD25 and CD69 (*Figure 1B*), and the CD8+ T cells started to express the cytotoxicity-related molecules FasL (*Krueger et al., 2003*) and lysosome-associated membrane protein 1 (LAMP1) (*Wolint et al., 2004*) (*Figure 1B*). These results indicate that CD8+ T cells contribute to the protective response to blood-stage malaria.

## CD8+ T cells contribute to protection in a FasL-dependent manner

The proportion of CD8+ T cells that express FasL increased after infection, suggesting that this molecule is involved in the immune response. To investigate this possibility, FasL-mutant *gld* mice were infected with PyNL. The course of infection in the *gld* mice resembled that in mice depleted of CD8+ T cells, insofar as parasitemia was exacerbated before peak parasitemia and the survival rate was lower than in wild-type (WT) mice (*Figure 1C*). Thus, FasL is important in controlling blood-stage malaria. Although we hypothesized that FasL expressed on CD8+ T cells is crucial, the FasL expressed on CD4+ T cells (*el-Khatib et al., 1995*; *Hahn et al., 1995*) may also play a protective role. However, this is unlikely because infection did not increase the expression of FasL on CD4+ T cells, in contrast to CD8+ T cells (*Figure 1D*). To confirm these inferences, we used cell transfer experiments combined with a prime–boost live vaccination system in which CD8+ and CD4+ T cells isolated from mice that had recovered from PyNL infection after two homologous boosts with PyL transferred protection from an otherwise lethal infection with PyL to the recipient mice (*Figure 1E,F*) (*Imai et al., 2010*). Mice that had received *gld* immune CD8+ T cells exhibited higher parasitemia at an early stage of infection, and some of them failed to control the challenge infection and died (*Figure 1F*, left panel). In contrast, CD4+ T cells from *gld* donors protected the recipients from challenge with PyL, similar to the protection afforded mice by CD4+ T cells from WT donors (*Figure 1F*, right panel). Therefore, FasL plays a crucial role in CD8+-T-cell-mediated protective immunity against blood-stage malaria.

To confirm the roles of CD8+ T cells responsible for resistance, we excluded the possibility that contaminants in transferred cells play a role using *Rag2*−/− mice as recipients (*Figure 1—figure supplement 2A*). CD8+ T cells obtained from 'immune' CD45.1-bearing C57BL/6 mice could transfer protection to *Rag2*−/− mice (CD45.2), although parasitemia was greater compared with when the irradiated mice were used as hosts (*Figure 1—figure supplement 2D*, *Imai et al., 2010*). In such lymphopenic recipients, a very few cells proliferate intensively and acquire effector capacities. Thus, the homeostatic proliferation of contaminants other than CD8+ T cells was evaluated. The ratio of the contaminated CD4+ cells in sorted CD8+ cells used for the transfer was less than 0.1% (*Figure 1—figure supplement 2B*). That in donor-derived CD45.1+ cells recovered from the recipients did not exceed 0.8%, and CD8+ T cells constantly occupied more than 96% even after infection with PyNL (*Figure 1—figure supplement 2C*). To further exclude the possibility, we determined how many CD4+ T cells are required to protect the recipient. The recipients transferred with $1 \times 10^7$ CD4+ T cells could control the challenge infection, but those with $1 \times 10^6$ CD4+ T cells could not (*Figure 1—figure supplement 2D*). Thus, 0.1% contaminated CD4+ T cells corresponding to $1 \times 10^4$ in transferred $10^7$ CD8+ T cells seem to have no protective ability. Based on these findings, we concluded that CD8+ T cells are responsible for the transferred protection.

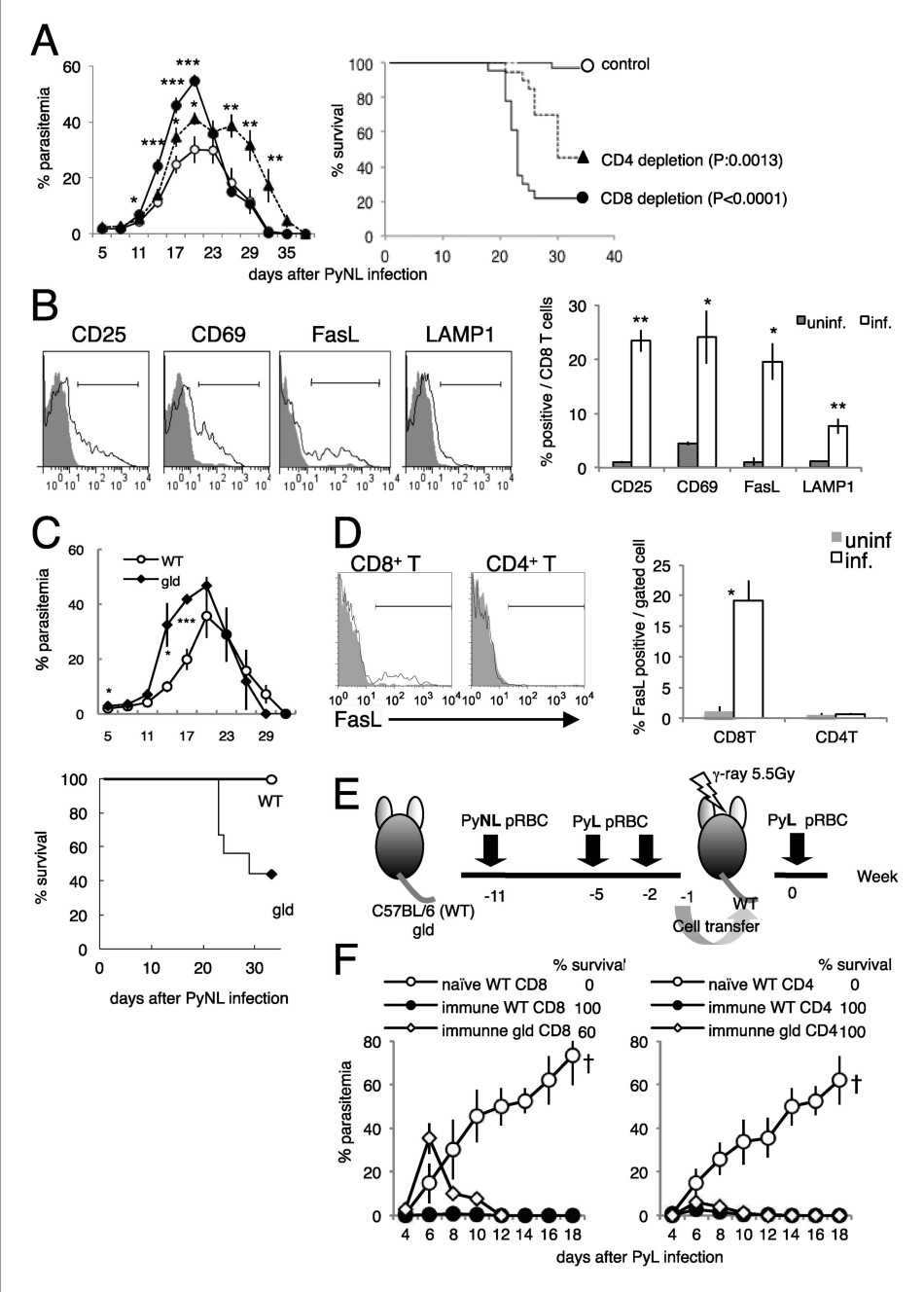

**Figure 1**. CD8[+] T cells and FasL protect against infection with *P. yoelii* NL (PyNL). (**A**) Daily parasitemia and survival rates of C57BL/6 mice depleted of CD8[+] or CD4[+] T cells after infection with PyNL. Parasitemia was estimated from microscopic observation of Giemsa-stained blood films. Parasitemia values are means ± SE of three pooled experiments (control: N = 20; CD4 depletion: N = 20; CD8 depletion: N = 23). *p < 0.05, **p < 0.01, and ***p < 0.001, Mann–Whitney U-test. Survival rate was calculated from three pooled individual experiments, as described above. p values for the Kaplan–Meier log rank test are shown. (**B**) Flow-cytometric analyses of splenic CD8[+] T cells were performed 7 days after infection with PyNL. Single-cell suspensions from spleens were stained with fluorescence-labeled anti-CD8β antibody. CD8[+] T cells were analyzed for the expression of the indicated molecules. Shaded areas and lines in the histograms represent their expression in uninfected and infected mice. Bar graph indicates percentages of CD8[+] T cells expressing the molecules as means ± SD of five mice from one of three experiments. *p < 0.05, **p < 0.01, Mann–Whitney U-test. (**C**) Daily parasitemia (upper panel) and survival rates (bottom panel) of wild type mice (WT) or *gld* mice infected with PyNL, monitored as in *Figure 1A*. WT, N = 6; *gld*, N = 9. Parasitemia values are means ± SD from one of five experiments. *p < 0.05 and ***p < 0.001, Mann–Whitney

*Figure 1. continued on next page*

*Figure 1. Continued*

U-test. Survival rates are from five pooled individual experiments (WT, N = 28; *gld*, N = 25). (**D**) Contribution of FasL expressed on CD8[+] T cells to the protective effects against blood-stage malaria. Expression of FasL on splenic CD4[+] T cells was evaluated. *p < 0.05, Mann–Whitney U-test. Data of FasL on CD8 are the same experiment as *Figure 1B*. (**E**) Experimental protocol for the adaptive transfer of cells after the prime–boost PyNL vaccine regime against lethal PyL infection. WT and *gld* mice were infected with PyNL, and then boosted twice with PyL. CD4[+] and CD8[+] T cells isolated from the vaccinated donors were transferred into irradiated recipients. Note that although some *gld* mice died from the PyNL infection, the survivors were as resistant to PyL infection as the WT mice. (**F**) Parasitemia was monitored in the recipients of the indicated cells. Each symbol indicates means ± SD. Each group contained five mice. The final survival rate of each group is also indicated. The results are from one experiment, representative of the two performed. Dagger indicates death.

The following figure supplements are available for figure 1:

**Figure supplement 1**. CD8[+] T cells play protective roles in C57BL/6 mice and BALB/c mice infected with PyNL.

**Figure supplement 2**. Confirmation that CD8[+] T cells are responsible for transferring protection to *Rag2*[−/−] mice.

## Malaria-parasite-infected erythroblasts express Fas

We next examined the cell types targeted by FasL-dependent immunity. FasL interacts with Fas expressed on target cells, inducing the apoptosis of the Fas-expressing cells (*Nagata and Golstein, 1995*). Recently, erythroid cells have been reported to express Fas (*De Maria et al., 1999*; *Tsushima et al., 1999*; *Mandal et al., 2005*; *Liu et al., 2006*). Based on our previous finding that malaria parasites infect erythroblasts (*Imai et al., 2013*). We postulated that infected erythroid cells are the targets of FasL-expressing CD8[+] T cells. Therefore, we analyzed the expression of Fas on infected erythroid cells in the spleens and peripheral blood of mice infected with PyNL–green fluorescent protein (GFP). Very few TER119[+] erythroid cells expressed Fas in the peripheral blood, even among the infected GFP[+] cells (*Figure 2*). In contrast, a number of infected GFP[+] cells expressing Fas were present in the spleen, and the frequency of those cells among the parasitized cells reached 50% before peak parasitemia (*Figure 2A,B*). To identify the erythroid cells that express Fas in the spleen, we examined the expression of MHC class I molecules on the infected cells because erythroblasts are distinguished from reticulocytes and mature RBCs by their high-level expression of MHC class I antigens (*Imai et al., 2013*). Almost all Fas-expressing cells, both infected and uninfected, were MHC class I[hi] (*Figure 2C*), indicating that the infected Fas[+] cells were erythroblasts. As those cells present antigens in conjunction with MHC class I molecules and are recognized antigen-specifically by CD8[+] T cells (*Imai et al., 2013*), it is possible that FasL-bearing CD8[+] T cells affect infected erythroblasts expressing Fas. Notably, the infection of erythroblasts with PyNL may induce their expression of Fas, because Fas[−] erythroblasts were markedly reduced in the infected cells relative to their numbers in uninfected cells (41% and 14%, respectively; *Figure 2C*). Moreover, the intensity of Fas expression was much higher on parasitized erythroblasts than in uninfected erythroblasts.

## CD8[+] T cells induce PS externalization on parasitized erythroblasts via FasL

As a consequence of the interaction between FasL and Fas, Fas-expressing cells undergo apoptosis (*Nagata, 1996a*, *1996b*), which is characterized by the fragmentation of their nuclei and the exposure of PS on the surface of the cell (*Yoshida et al., 2005*). PS-displaying infected RBCs are more susceptible to phagocytosis, and this phenomenon is involved in the protection of the host from malaria. Therefore, we investigated whether PS is exposed on erythroid cells in response to the FasL–Fas interaction during malaria (*Figure 3*). PS[+] cells were significantly increased in splenic infected TER119[+] cells (*Figure 3A*). CD8[+]-T-cell-depleted or *gld* mice had much fewer PS[+] cells than the control mice (*Figure 3B,C*). Moreover, the majority of infected Fas[+] splenic erythroblasts displayed PS (*Figure 3D*), suggesting that CD8[+] T cells and FasL are involved in increasing the exposure of PS on infected cells in the spleen. In contrast, the number of PS[+] cells among the infected RBCs was only slightly increased in the peripheral blood. Because the *gld* and CD8[+]-T-cell-depleted mice contained

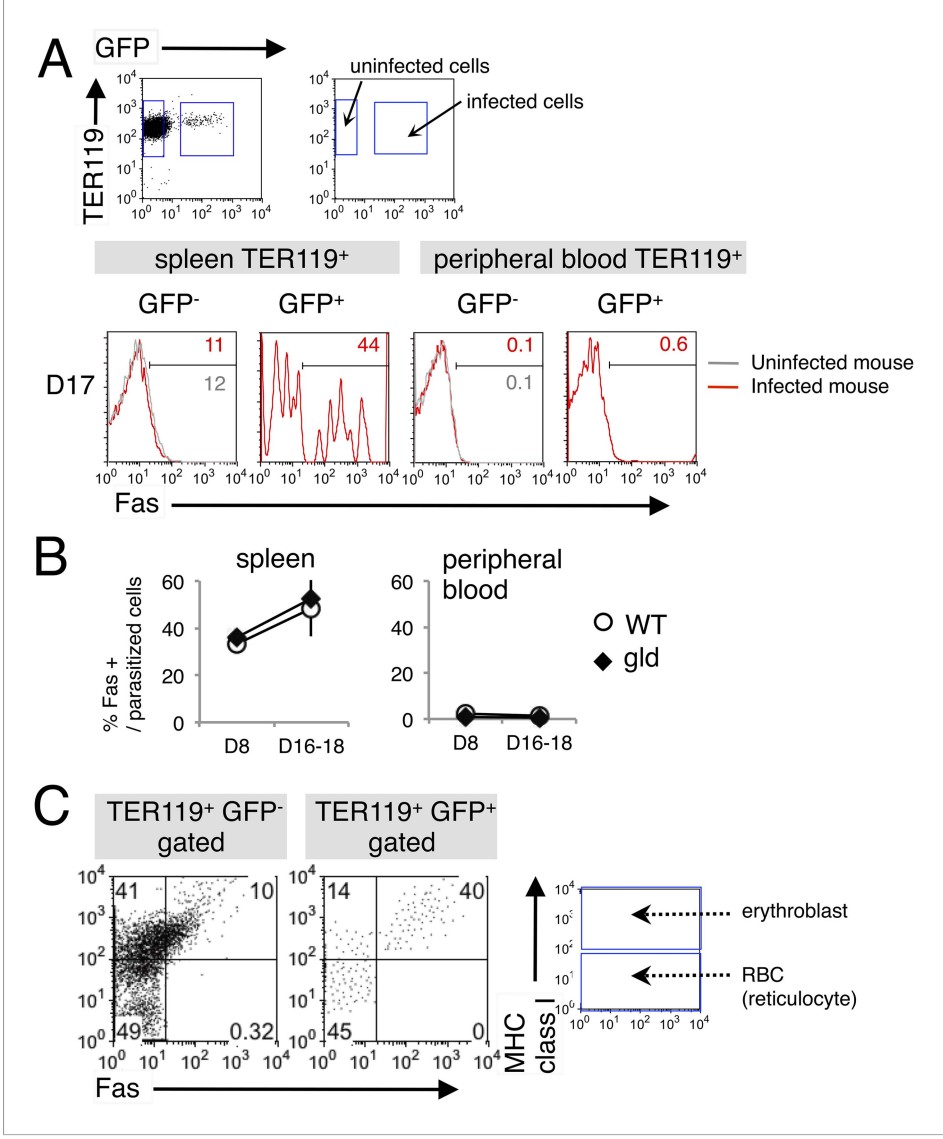

**Figure 2.** Fas is expressed on erythroid cells infected with PyNL. (**A**) Spleen cells and peripheral blood cells obtained from mice infected with PyNL–GFP were stained with anti-TER119, anti-Fas, and anti-MHC class I antibodies. TER119+ GFP+ infected or TER119+ GFP− uninfected cells were analyzed for their expression of Fas. Numbers on the histograms indicate the percentages of Fas+ cells in the gated cells. (**B**) Percentages of Fas+ cells in parasitized cells (TER119+ GFP+ Fas+/TER119+ GFP+) are shown as means ± SD from one experiment (N = 4), representative of the three performed. (**C**) Splenic TER119+ cells infected (right panel) or uninfected (left panel) in mice infected with PyNL–GFP were separated into MHC class I$^{hi}$ erythroblasts (fluorescence intensity > 10$^2$), class I$^{lo-neg}$ reticulocytes, and mature RBCs and analyzed for their Fas expression. Numbers indicate the percentages of the gated cells in each quadrant.

fewer PS+ infected RBCs, the increase in PS+ cells seemed to be dependent on FasL and CD8+ T cells, despite the absence of Fas+ cells in the peripheral blood.

To further investigate the involvement of CD8+ T cells in PS exposure, splenic TER119+ cells isolated from *gld* mice, which contained fewer PS+ cells despite similar numbers of Fas+ cells (***Figures 2B, 3C***), were cocultured with CD8+ T cells of various origins (***Figure 4A***). CD8+ T cells from infected WT mice efficiently induced PS exposure in a dose-dependent manner, whereas those from uninfected WT mice did not (***Figure 4B***). Exposure of PS was only observed in infected GFP+ cells, and not in uninfected cells (***Figure 4C***). Importantly, CD8+ T cells from infected *gld* mice induced PS

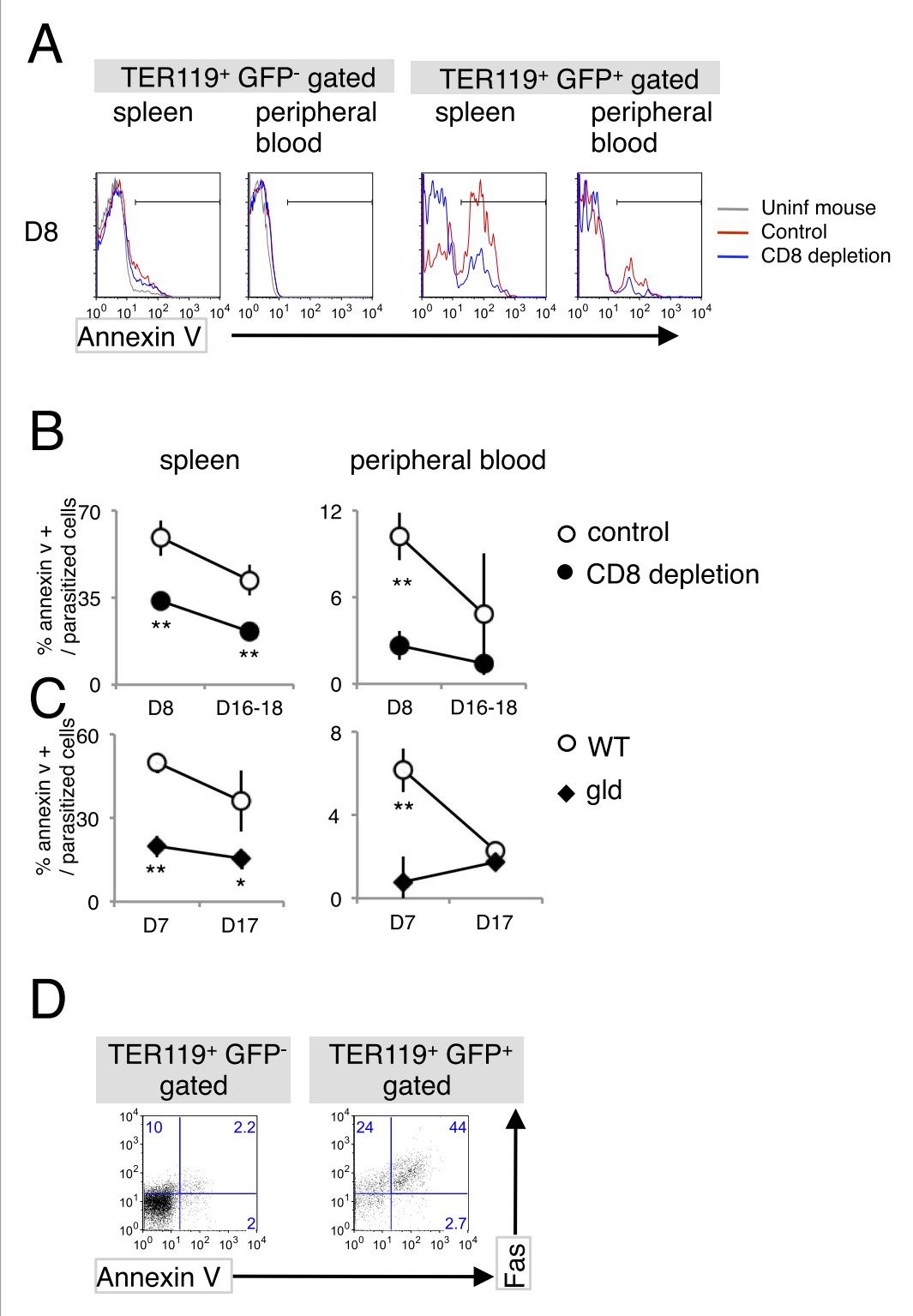

**Figure 3**. Infection with PyNL induces externalization of phosphatidylserine (PS) on parasitized cells. (**A**) Spleen cells and peripheral blood obtained from the indicated mice 8 days after infection with PyNL–GFP were stained with anti-TER119 antibody and annexin V. Infected GFP+ or uninfected GFP− TER119+ cells were analyzed for the expression of PS. (**B**) Percentages of TER119+ GFP+ PS+ cells in the TER119+GFP+ cells in the control (open symbols) and CD8+-depleted mice (closed symbols) are shown as means ± SD from one experiment (N = 4), representative of the three performed. (**C**) Those in the *gld* mice were also analyzed. \*\*p < 0.01, Mann–Whitney U-test. (**D**) Splenic TER119+ cells

*Figure 3. continued on next page*

*Figure 3. Continued*
infected (right panel) or uninfected (left panel) obtained from mice 8 days after infection with PyNL–GFP were analyzed for the expression of PS and Fas. Numbers indicate the percentages of the gated cells in each quadrant.

exposure in only a few infected GFP$^+$ cells, as did the CD8$^+$ T cells from uninfected WT mice (*Figure 4B,C*), indicating that FasL expressed by CD8$^+$ T cells is responsible for PS exposure.

Next, we examined FasL-dependent PS exposure in vitro using FasL–Strep. Splenic TER119$^+$ cells collected from *gld* mice infected with PyNL–GFP were cultured with FasL–Strep (*Figure 4D*). The addition of FasL–Strep induced the externalization of PS in infected GFP$^+$ cells in a dose-dependent manner (*Figure 4E*). Because the Fas-expressing cells in the spleen were erythroblasts (*Figure 2*), we confirmed that FasL–Strep induced erythroblasts to expose PS. The PS$^+$ cells were all MHC class I$^{hi}$ erythroblasts, even in the absence of FasL–Strep (*Figure 4F*). Furthermore, FasL–Strep reduced the proportion of MHC class I$^{hi}$ erythroblasts in PS$^-$cells (from 64% to 50%), indicating that PS$^-$ Fas-expressing erythroblasts expose PS in the presence of FasL–Strep (*Figure 4F*). In contrast, PS exposure was not observed on infected cells from the peripheral blood, even in the presence of CD8$^+$ T cells or FasL–Strep (*Figure 5*), presumably because Fas was not expressed on these cells (*Figure 2*). FasL is known to localize to the cell surface, but it is also secreted (*Morello et al., 2013*). Therefore, we determined whether FasL-dependent PS exposure requires cell contact or a soluble factor. Transwell experiments revealed that PS was only displayed on infected erythroid cells when the erythrocytes contacted CD8$^+$ T cells, and was not affected by soluble factors secreted from CD8$^+$ T cells (*Figure 6*). These results indicate that activated CD8$^+$ T cells can induce PS exposure on infected splenic cells (parasitized erythroblasts) in a FasL- and contact-dependent manner during blood-stage malaria, although the involvement of exosomes bearing pro-apoptotic membranous FasL could not be ruled out (*Andreola et al., 2002*).

## PS exposure enhances phagocytosis by macrophages

In line with previous reports from ourselves and others on the importance of the phagocytosis of infected RBCs in the protective response to blood-stage malaria (*Zhang et al., 1999*; *Couper et al., 2007*; *Imai et al., 2010*; *Matsuzaki-Moriya et al., 2011*; *Duan et al., 2013*), mice depleted of macrophages by administration with clodronate/liposome (C/L) quickly died in association with high parasitemia (*Figure 7A,B*). It has also been reported that PS-exposing cells are highly susceptible to phagocytosis by macrophages (*Fadok et al., 1998*; *van den Eijnde et al., 1998*). These findings led us to hypothesize that PS exposure on infected cells, induced by CD8$^+$ T cells, accelerates their engulfment by phagocytes such as macrophages. To test this hypothesis in vitro, infected cells were isolated from CD8$^+$-T-cell-deleted, *gld*, and WT control mice infected with PyNL. The degree of PS exposure was different in each preparation. Those cells were then labeled and cocultured with macrophages from uninfected mice (*Figure 7C*). Only a few macrophages phagocytosed the uninfected cells, whereas up to 30% of macrophages engulfed the infected cells from the WT mice. When cocultured with infected cells from CD8$^+$-T-cell-depleted WT or *gld* mice, the number of phagocytotic macrophages was significantly reduced (*Figure 7D*). A positive correlation between the degree of PS exposure on the infected cells and the numbers of phagocytotic macrophages was observed, indicating that PS$^+$ cells are readily phagocytosed (*Figure 7E*).

The phagocytosis of the infected cells was analyzed in vivo using PyNL–GFP. Spleen and peripheral blood cells were stained with CD11b and separated into CD11b$^+$ GFP$^+$ cells and CD11b$^-$ GFP$^+$ cells, as infected cells phagocytosed by macrophages or infected cells, respectively (*Figure 8A,B*). Macrophages phagocytosing infected GFP$^+$ cells were observed under a microscope (*Figure 8C*). There were more infected CD11b$^-$ GFP$^+$ cells in the peripheral blood of CD8$^+$-T-cell-depleted mice than in the peripheral blood of the control mice (*Figure 8A*), reflecting the higher parasitemia in the CD8$^+$-T-cell-depleted mice (*Figure 1A*). Substantial numbers of infected cells were phagocytosed in the spleen but not in the peripheral blood, indicating the importance of this organ in the elimination of the malaria parasite (*Figure 8A*). The proportion of phagocytosed infected cells in the total infected cells (CD11b$^+$ GFP$^+$/GFP$^+$) in the spleens of CD8$^+$-T-cell-depleted mice was significantly lower than the proportion in the control mice (*Figure 8A,B,D*). Furthermore, *gld* mice showed similar results to

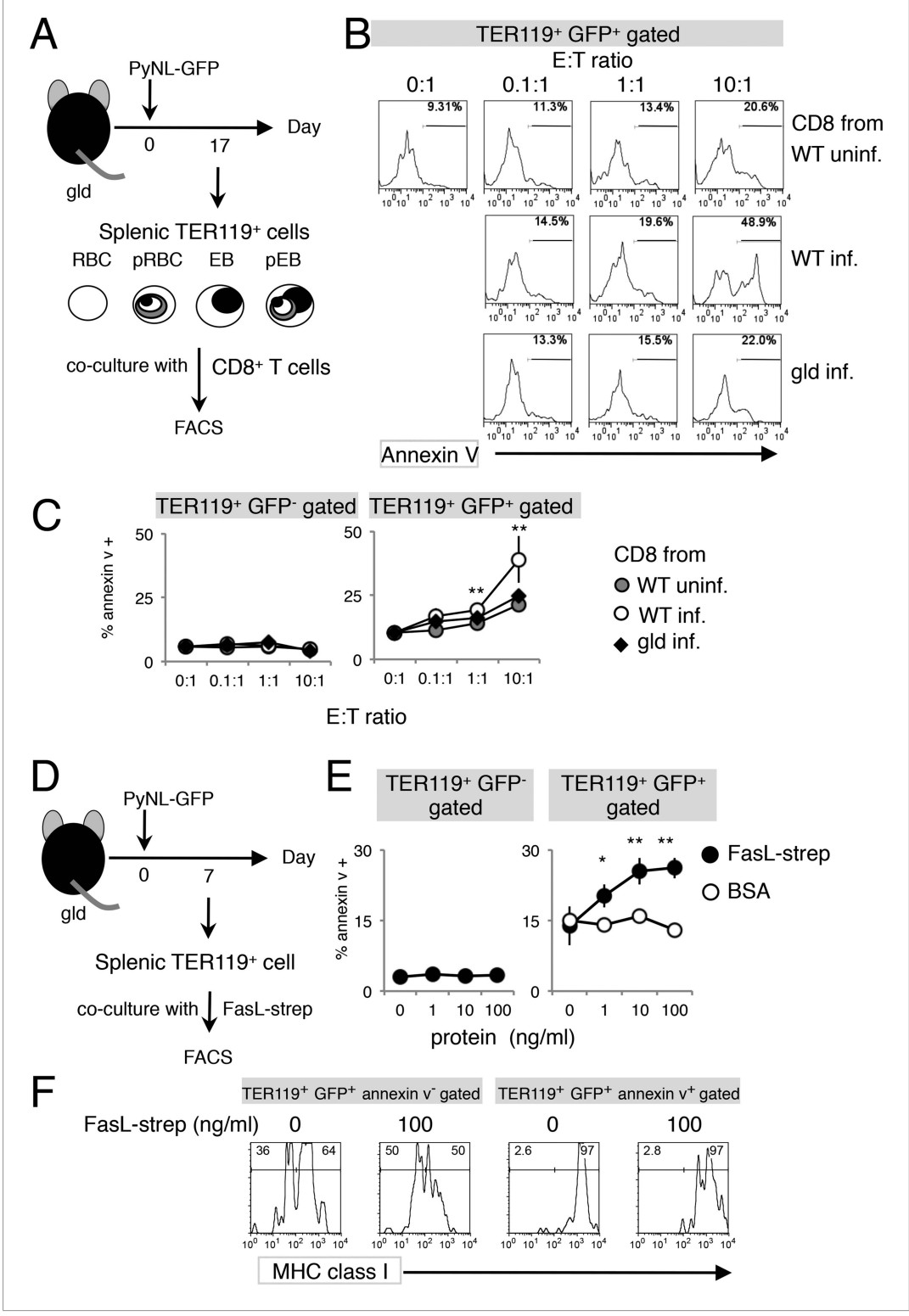

**Figure 4**. Exposure of PS is dependent on CD8+ T cells and FasL. (**A**) Experimental protocol for the evaluation of CD8+-T-cell-dependent PS externalization in parasitized cells in vitro. Splenic TER119+ cells containing RBC, pRBC, erythroblasts (EB) and pEB (3 × 10^5) isolated from *gld* mice 17 days after infection with PyNL–GFP were cultured for 4 hr with CD8+ T cells from WT or *gld* mice 17 days after PyNL infection, at the indicated ratios. (**B**) Cultured TER119+ cells with CD8+ T cells from the indicated mice were stained with annexin V, and GFP+ cells were analyzed for PS expression. Numbers in histograms indicate percentages of annexin V+ cells in the gated cells. (**C**) Values are

*Figure 4. continued on next page*

*Figure 4. Continued*

means ± SD from triplicate cultures in one experiment, representative of the four performed. **p < 0.01, Mann–Whitney U-test. (**D**) Experimental protocol for the evaluation of FasL-dependent PS externalization in parasitized cells in vitro. TER119+ cells isolated from spleens and peripheral blood of *gld* mice 7 days after infection with PyNL–GFP were cultured for 4 hr with the indicated amounts of FasL–*Strep* or bovine serum albumin (negative control). (**E**) Cultured cells were collected and stained with annexin V, and annexin V+ cells among the GFP+ parasitized cells were quantified. Values are means ± SD of triplicate cultures in one experiment, representative of the four performed. *p < 0.05 and **p < 0.01, Mann–Whitney U-test. (**F**) Annexin V-positive or -negative GFP+ parasitized cells were analyzed for the expression of MHC class I antigens, as in *Figure 3C*.

those of CD8+-T-cell-depleted mice (*Figure 8E*). Finally, we analyzed macrophage subsets and found that F4/80+ red pulp macrophages are responsible for the ingestion of parasites. SIGNR1+ marginal zone macrophages, CD169+ marginal metallophilic macrophages, and CD68+ tingible-body macrophages appeared not to be involved in phagocytosis (*Figure 8F*). Although depletion of CD8+ T cells did not affect the numbers of each macrophage subset (data not shown), it dramatically reduced the number of phagocytic F4/80 macrophages.

As the macrophages in the CD8+-T-cell-depleted mice were activated to a similar degree as those in the control mice during malaria (*Figure 9*), the proportion of cells exposing PS may correspond to this difference in the number of phagocytosing macrophages. These results indicate that the phagocytosis of infected cells occurs in the spleen and correlates with the exposure of PS on the infected cells, which is dependent on CD8+ T cells and FasL. We obtained the same results using dendritic cells instead of macrophages (*Figure 8—figure supplement 1*).

## Macrophages phagocytose infected cells via Tim-4

Recently, T-cell immunoglobulin- and mucin-domain-containing molecule (Tim-4; also known as Timd4) was identified as a PS receptor (*Miyanishi et al., 2007*). In this study, the phagocytosis of PS-exposing infected erythroid cells was observed. Therefore, we investigated the involvement of Tim-4 as a novel receptor in the protective immune response against malaria. The expression of Tim-4 on splenic macrophages was upregulated, and the number of Tim-4+ macrophages increased in response to infection with PyNL (*Figure 10A*). The phagocytosis by macrophages of infected RBCs isolated from infected WT mice was dose-dependently inhibited by the presence of antibodies directed against Tim-4 (*Figure 10B,C*). These results indicate that Tim-4 contributes to the phagocytosis of infected RBCs.

## Discussion

Here, we have demonstrated a novel protective mechanism against blood-stage malaria conferred by CD8+ T cells. CD8+ T cells interact with infected erythroblasts and induce them to display PS in a FasL-dependent manner. In turn, PS exposure enhances the susceptibility of infected cells to phagocytosis, which contributes to the elimination of the parasite. Our proposal may resolve the controversial protective roles of CD8+ T cells against infected erythroid cells.

Vinetz et al. had reported that CD8+ T cells are not contributed to protection against blood-stage murine malaria (*Vinetz et al., 1990*). They used *P. yoelii* 17X clone 1.1, which results in an obviously different course of infection from ours. The PyNL clone that we used appears more virulent than the 17× clone 1.1 as judged by the higher peak parasitemia (30–40% vs 10%) and prolonged period for parasite elimination (30 days vs 15 days), suggesting that the difference in virulence may cause the different results when mice were depleted of CD8+ T cells.

It is quite possible that CD8+ T cells target erythroblasts that strongly express MHC class I antigens. However, we previously reported the contribution of macrophages to CD8+-T-cell-mediated protection against malaria (*Imai et al., 2010*). Those findings, together with the present study, suggest that CD8+ T cells enhance not only the phagocytotic capacity of macrophages but also the susceptibility of infected erythroblasts to phagocytosis through their display of PS. Thus, this FasL-dependent effect of CD8+ T cells on infected erythroblasts might be essential for the protective immune response to blood-stage malaria by supporting enhanced phagocytosis. Thus, CD8+ cells collaborate with macrophages to completely eradicate the parasites.

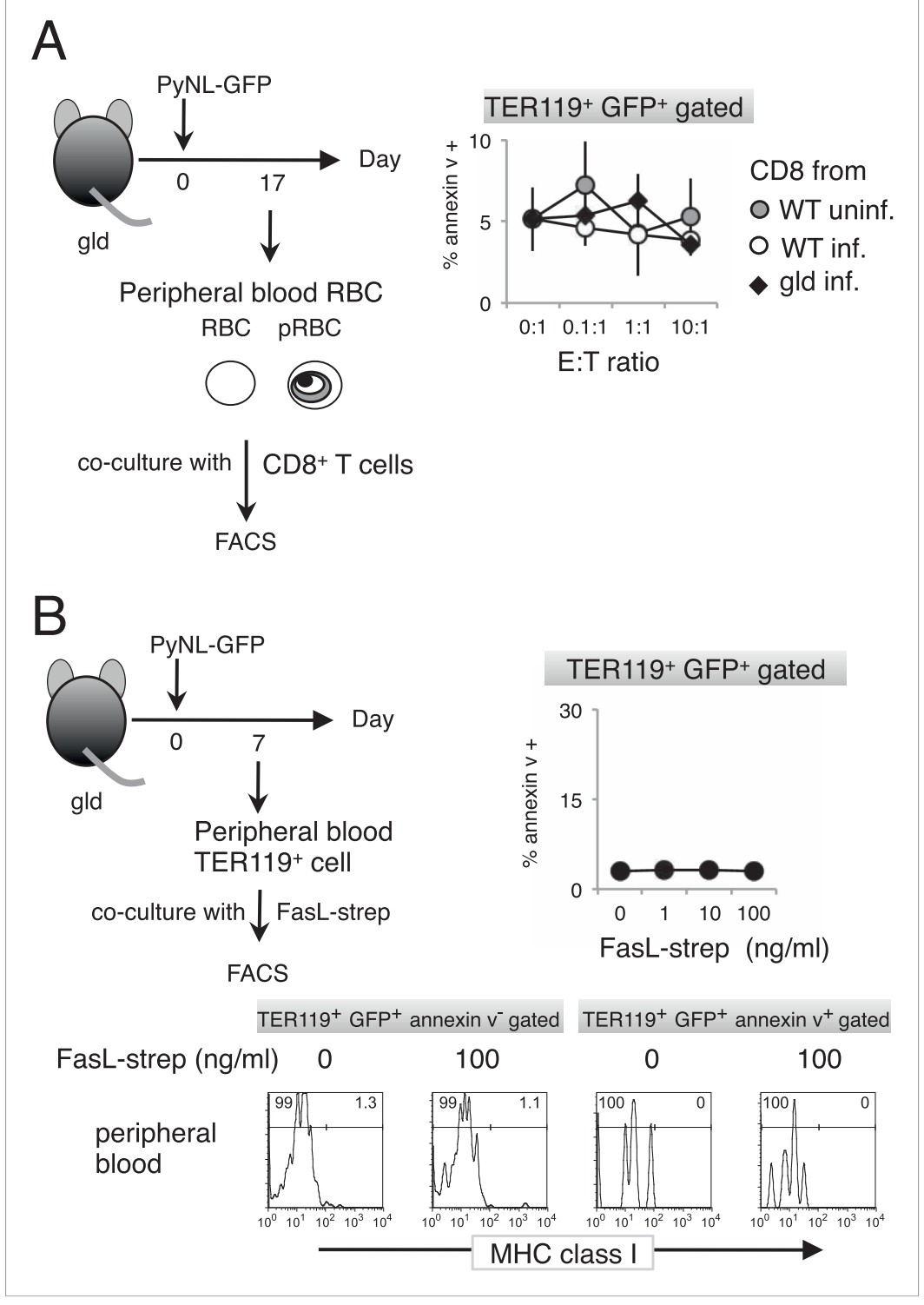

**Figure 5**. Externalization of PS in pRBCs was not induced in vitro. Peripheral blood cells obtained from *gld* mice infected with PyNL–GFP were cultured with CD8+ T cells (**A**) or FasL–*Strep* (**B**) and analyzed as in *Figure 4*.

The CD8+-T-cell-mediated protection that targets parasitized erythroblasts may operate in the early phase of infection, as inferred from the course of infection in mice depleted of CD8+ T cells. We have previously shown that the proportion of infected erythroblasts is constant during the course of

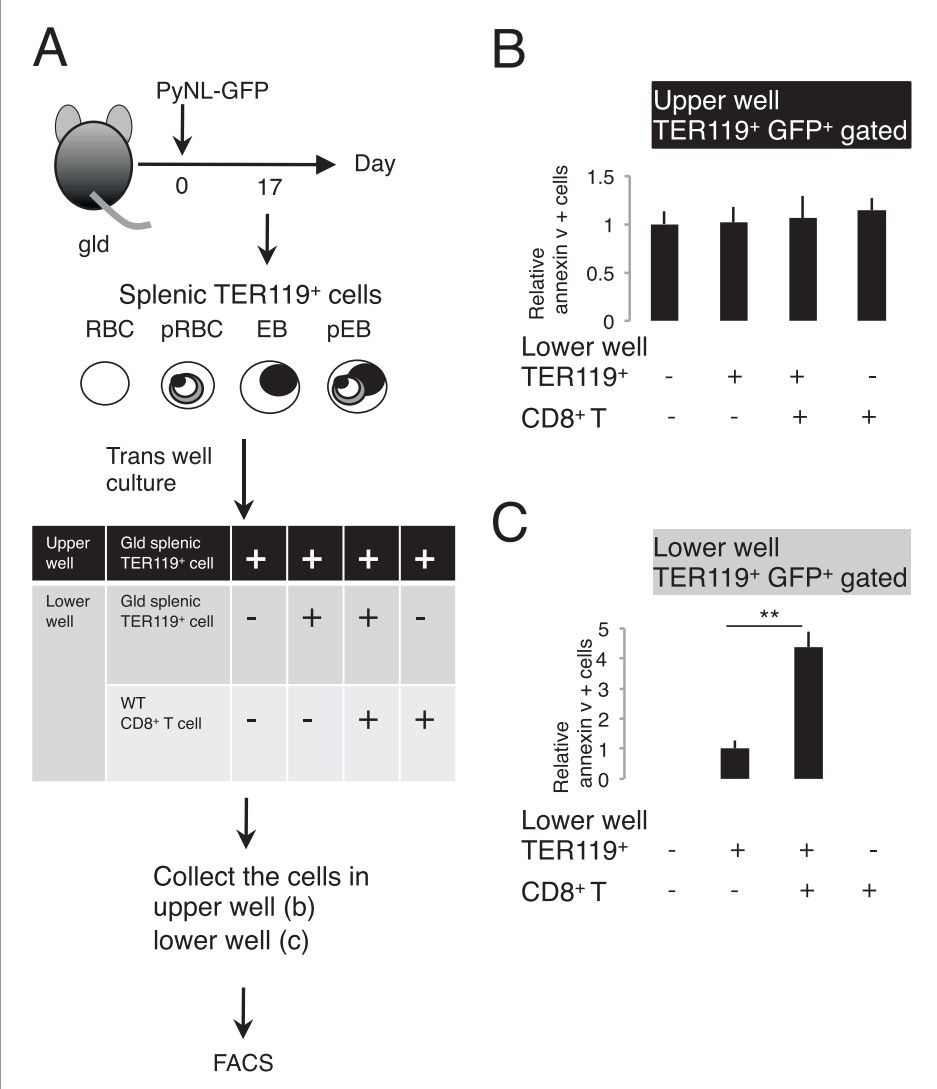

**Figure 6**. Externalization of PS in parasitized cells requires contact with CD8+ T cells. (**A**) Protocol of the contact dependence assay using Transwell cultures. Splenic TER119+ cells from *gld* mice infected with PyNL–GFP and CD8+ T cells from WT mice infected with PyNL were placed into the upper and/or lower wells and cultured for 6 hr. GFP+ parasitized cells were analyzed for PS expression, as in *Figure 4B*. The ratio of the percentages of PS+ cells in the GFP+ cells in the upper (**B**) and lower wells (**C**) was calculated as (% PS+ GFP+ of GFP+ cells in each test)/(% PS+GFP+ in GFP+ cells in the absence of cell components in the lower well) in (**B**), and as (% PS+ GFP+ in GFP+ cells in the presence of CD8+ T cells)/(% PS+ GFP+ in GFP+ cells in the absence of CD8+ T cells in the lower well) in (**C**). Values shown are the means ± SD of triplicate cultures in one experiment, representative of the three performed. **p < 0.01, Mann–Whitney U-test.

infection, unlike the proportion of infected RBCs, which increases dramatically in the later stages of infection (*Imai et al., 2013*). This means that there are relatively more infected erythroblasts in the early stage of infection. Therefore, the reduction of infected erythroblasts by CD8+ T cells in the early phase would efficiently control blood-stage malaria. From this perspective, this protective mechanism might effectively control malaria parasites in humans, in which parasitemia develops to a lower level than that observed in animal models. Indeed, parasitized erythroblasts were found within the bone marrow of patients with vivax malaria (*Ru et al., 2009*), and *P. falciparum* parasites (*Tamez et al., 2009*) can infect erythroblasts in vitro. Therefore, these cells might be targets of CD8+ T cells in

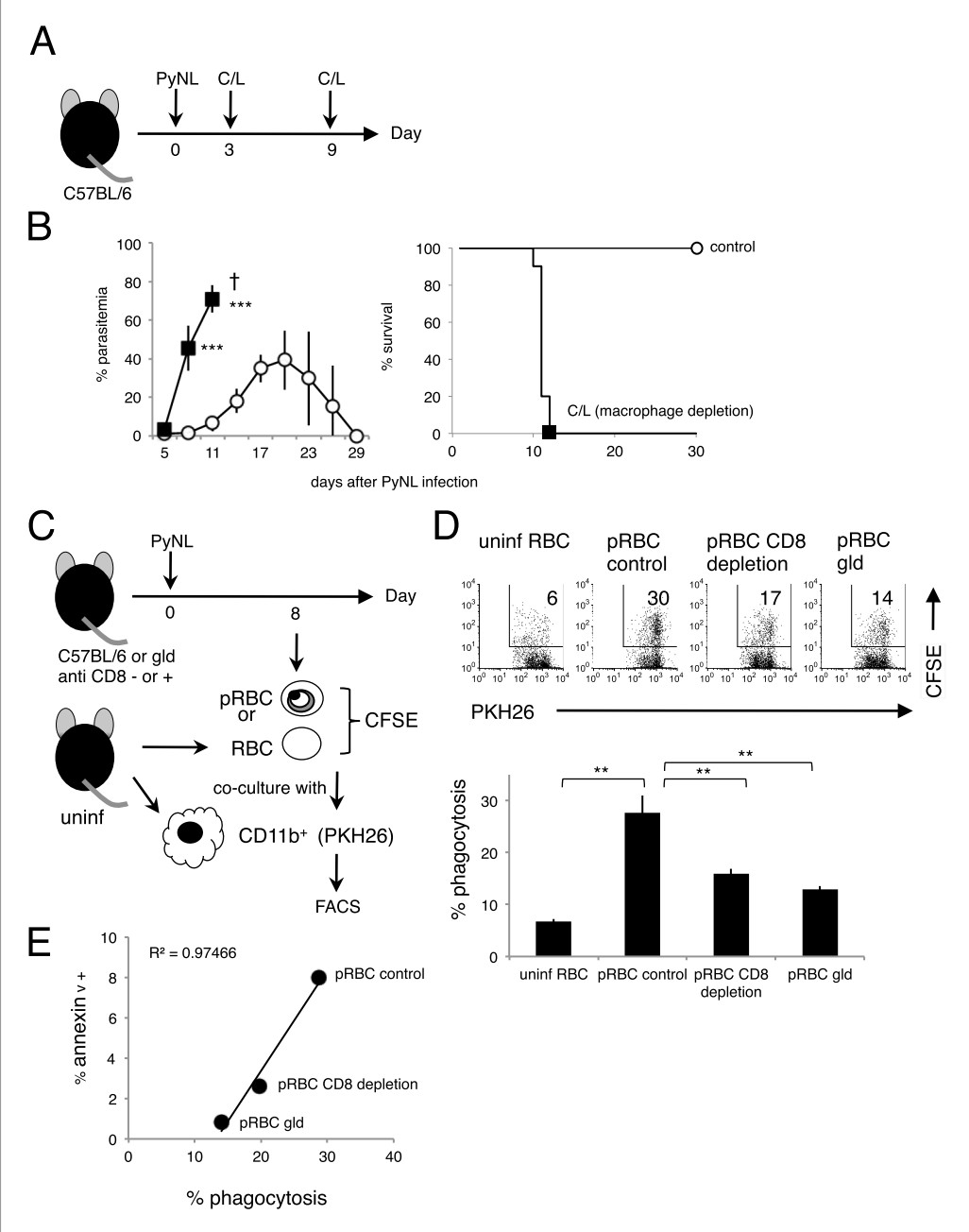

**Figure 7**. Phagocytosis of parasitized RBCs (pRBCs) by macrophages correlates with RBC PS expression in vitro. (**A**) Experimental protocol for depleting macrophage with clodronate/liposomes (C/L). (**B**) Parasitemia (left panel) and survival rate (right panel) were evaluated from two pooled separate experiments. Control: N = 17; C/L: N = 10. ***p < 0.001, Mann–Whitney U-test. (**C**) Protocol used to evaluate the phagocytosis of pRBCs. pRBCs obtained from WT, CD8[+]-depleted, or *gld* mice were labeled with CFSE, and then cocultured for 4 hr with CD11b[+] macrophages obtained from uninfected WT mice, which had been labeled with PKH26 fluorescence. RBC from uninfected WT mice was also tested. The ratio of macrophages to pRBCs or RBC was 1:30. (**D**) Phagocytosis was evaluated by detecting the PKH[+] CFSE[+] macrophages after culture with pRBCs isolated from the indicated mice. Numbers in the upper panels indicate the percentage of phagocytic macrophages in the squares in the total macrophages (% phagocytosis = PKH[+] CFSE[+]/PKH[+]). Values in the bar graph are means ± SD from triplicate cultures in one experiment, representative of the two experiments performed. **p < 0.01, Mann–Whitney U-test. (**E**) PS exposure correlates with the degree of phagocytosis. The percentage of PS[+] cells in each pRBC preparation and the percentage of phagocytic macrophages when each preparation was used are plotted.

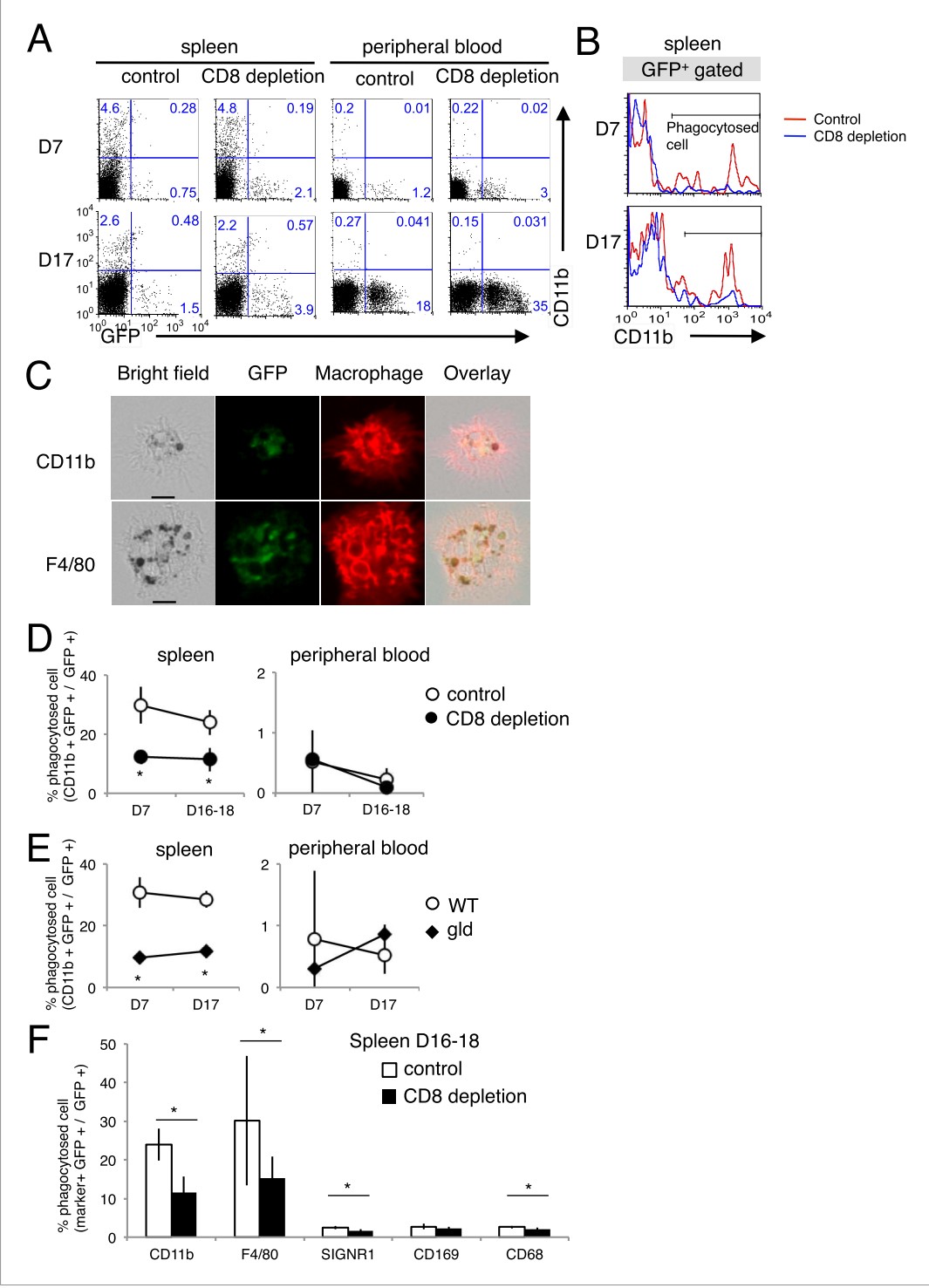

**Figure 8**. Phagocytosis of parasitized cells by macrophages in vivo. Spleen cells and peripheral blood were isolated seven or 17 day after mice depleted of CD8+ T cells were infected with PyNL–GFP. (**A**) Those cells were then stained with anti-CD11b antibody and separated into free parasitized cells (GFP+ CD11b−) and phagocytosed cells (GFP+ CD11b+). The numbers represent the percentages of cells in each quadrant. (**B**) Histograms indicate CD11b expression in GFP+ gated cells. (**C**) Images of phagocytosed parasitized cells are shown. Hemozoin-containing adherent macrophages were isolated from spleens with magnetic sorting and were observed microscopically. Scale bars represent 10 μm. Fractions of phagocytosed GFP+ parasitized cells were quantified as CD11b+ cells from mice depleted of CD8+ T cells (**D**) or from *gld* mice (**E**). (**F**) Macrophage subsets expressing the

*Figure 8. continued on next page*

*Figure 8. Continued*

indicated macrophage markers were also calculated in control and CD8+ T cell-depleted mice. Values are means ±
SD of 5–7 mice from three pooled individual experiments. *p < 0.05, Mann–Whitney U-test.

The following figure supplement is available for figure 8:

**Figure supplement 1**. Dendritic cells also phagocytose parasitized cells, presumably in response to PS exposure.

humans. It will be very interesting to evaluate whether parasitized erythroblasts are phagocytosed in the human bone marrow or spleen (although this will be difficult to demonstrate experimentally).

Notably, we have demonstrated the expression of Fas on erythroblasts infected with the malaria parasite (*Figure 2C*). There are two possible explanations for the expression of Fas on infected erythroblasts. One is that the infection of Fas⁻ erythroblasts with the malaria parasite induces the expression of Fas. Our findings support this because Fas⁻ erythroblasts were markedly reduced in infected cells compared with their numbers in uninfected cells, indicating a transition from Fas⁻ to Fas⁺ cells upon infection. However, the precise mechanism of this induction of Fas remains to be clarified. The other possible explanation is that the malaria parasite infects Fas⁺ erythroblasts. Erythroblasts are known to express Fas under physiological conditions, and Fas is considered to be involved in the negative regulation of erythropoiesis (*De Maria et al., 1999*; *Liu et al., 2006*). Furthermore, uninfected Fas⁺ erythroblasts were found in mice infected with PyNL. Activated CD8⁺ T cells expressing FasL might interact with uninfected erythroblasts expressing Fas and induce bystander cell damage during infection, and this system may underlie the pathogenicity of malarial anemia.

The Fas/FasL-dependent, CD8⁺ T cell-induced PS externalization by parasitized erythroblasts required cell-to-cell contact (*Figure 6*). Do CD8⁺ T cells contact their target cells in the spleen? As CD8⁺ T cells and erythroblasts occur in the splenic white pulp and red pulp, respectively (*Mebius and Kraal, 2005*), the opportunity for contact between them might be limited. However, infection with the malaria parasite changes the structure of the spleen and makes the white pulp indistinguishable from the red pulp (*Del Portillo et al., 2012*). Thus, it is possible that CD8⁺ T cells contact erythroblasts in the spleens of mice infected with PyNL. In vivo imaging would be useful in confirming this, because imaging has recently shown that CD8⁺ T cells accumulate in the liver after sporozoite infection (*Kimura et al., 2013*).

We did not investigate whether erythroblasts undergo apoptosis after the ligation of Fas, as in normal cells, and whether apoptosis suppresses parasite growth. Unlike viruses, malaria parasites can multiply inside RBCs but may not require any cellular machinery for their replication, suggesting that the apoptosis of the host cells may not influence parasite growth. Indeed, a related protozoan, *Toxoplasma gondii*, can survive inside damaged host cells (*Yamashita et al., 1998*). Instead, the elimination of the malaria parasite may require the phagocytosis of the infected cells by the phagocytes of the reticuloendothelial system, and the externalization of PS on parasitized erythroblasts via Fas/FasL plays an important role in this process. PS acts as an 'eat me' signal for phagocytes (*Savill and Gregory, 2007*) and contributes to the rapid removal of infected erythroblasts and apoptotic cells. Erythroblasts

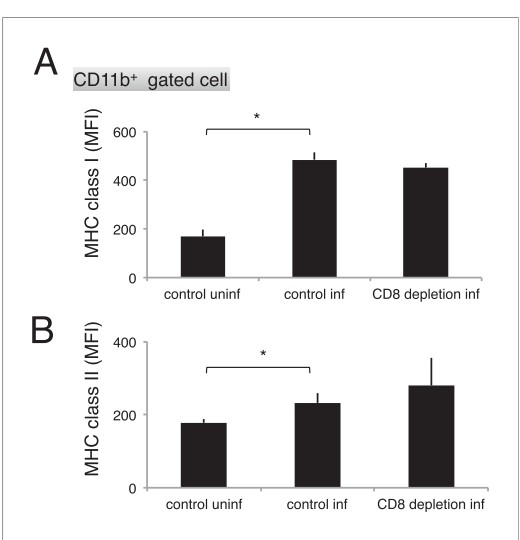

**Figure 9**. Depletion of CD8⁺ cells did not affect the activation of macrophages. Spleen cells collected from the indicated mice 17 days after infection with PyNL were stained with anti-CD11b, anti-MHC class I, and anti-MHC class II antibodies. CD11b⁺ cells were analyzed for their expression of class I (**A**) and class II molecules (**B**). Values shown are the means ± SD of 3–5 mice in one experiment, representative of the two performed. *p < 0.05, Mann–Whitney U-test.

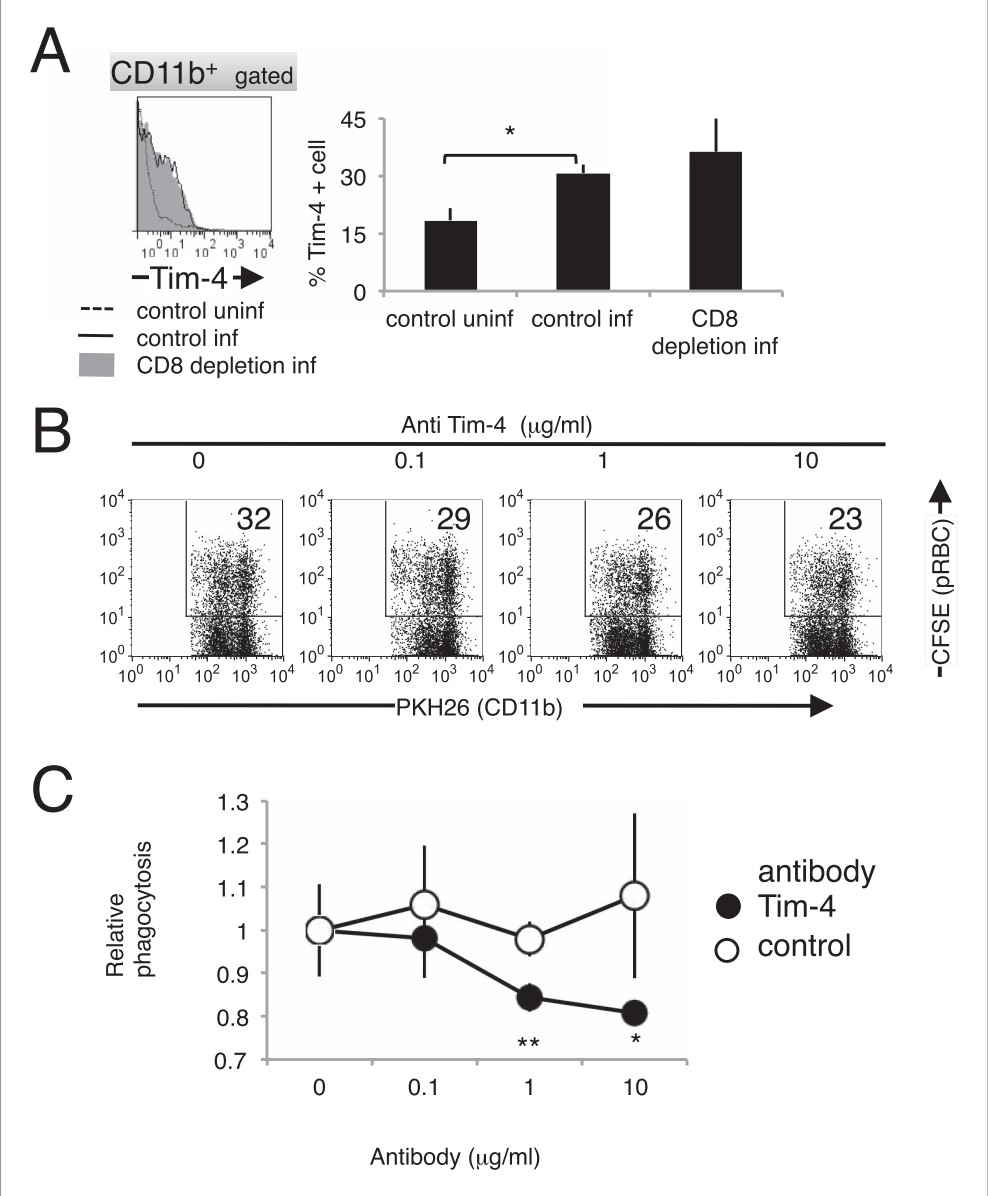

**Figure 10**. Tim-4 expressed on macrophages contributes to the phagocytosis of pRBCs. (**A**) Infection with PyNL induced the expression of Tim-4 on macrophages. Spleen cells obtained from mice 17 days after infection were stained with anti-CD11b and anti-Tim-4 antibodies, and the CD11b[+] cells were analyzed for Tim-4 expression. The expression levels of Tim-4 are shown in a histogram. Values in the bar graph are the means ± SD of six mice in two pooled individual experiments. *$p < 0.05$, Mann–Whitney U-test. (**B**) Addition of anti-Tim-4 antibody suppressed the phagocytosis of pRBCs by macrophages. PKH-labeled macrophages from uninfected mice were cultured with pRBCs isolated from WT mice 17 days after infection in the presence of the indicted concentrations of anti-Tim-4 antibody. The phagocytic macrophages were evaluated as in *Figure 7D*. (**C**) Inhibitory effects of anti-Tim-4 antibody on phagocytic cells were quantified as (% phagocytic macrophages in the presence of the antibody)/(% phagocytic macrophages in the absence of the antibody). Values are the means ± SD of triplicate cultures in one experiment, representative of the four performed. *$p < 0.05$ and **$p < 0.01$, Mann–Whitney U-test.

are distributed in a specific region called the 'erythroblastic island' in the splenic red pulp. Macrophages are located in the center of the erythroblastic island and rapidly phagocytose the nuclei of erythroblasts after their enucleation under physiological conditions (*Chasis and Mohandas, 2008*).

These macrophages may rapidly engulf infected erythroblasts as soon as PS is exposed after their interaction with CD8+ T cells.

Not only the erythroblasts in the spleen, but also the infected RBCs in the peripheral blood, expose PS in response to CD8+ T cells and FasL (*Figure 3*). Although PS exposure on infected RBCs induced by Fas stimulation could not be reproduced in vitro coincident with the absence of Fas+ cells in the peripheral blood, we observed a substantial number of infected PS+ RBCs in the peripheral blood. One possible explanation for FasL-dependent PS exposure on infected Fas− RBCs is that infected erythroblasts exposing PS develop into RBCs after enucleation, which is associated with the shedding of MHC class I molecules. PS exposure on infected RBCs has been reported in response to several stressors during malaria (*Foller et al., 2009*), and the FasL- and CD8+-T-cell-dependent system might be one cause of this PS exposure. PS exposure on infected RBCs might be part of the CD8+-T-cell-mediated protective mechanism against blood-stage malaria.

We proposed that Tim-4 is a novel phagocytic receptor for infected cells. The rate at which an anti-Tim-4 antibody inhibited the phagocytosis of infected RBCs (up to 20%) seems appropriate because 15–20% of the macrophages used here (obtained from uninfected mice) expressed Tim-4 (*Figure 10*). However, infection with PyNL induced the expression of Tim-4 on macrophages, which may play a major role in the phagocytosis of infected cells during malarial infection. Our results also indicate that other molecules that are known PS receptors, such as PS receptor (*Hoffmann et al., 2001*) and developmental endothelial locus 1 (Del-1) (*Hanayama et al., 2004*), might be involved in the phagocytosis of infected cells.

In summary, we have clearly demonstrated the protective mechanisms of CD8+ T cells against blood-stage malaria. Our findings should provide novel strategies for the development of a blood-stage vaccine based on the activation of CD8+ T cells, distinct from those strategies based on the induction of antibodies. Antigens recognized by antibodies must be expressed on the parasite's surface. Such molecules are exposed to immune pressure and acquire polymorphisms, allowing them to evade antibody recognition and causing 'strain-specific immunity', which hampers the development of effective vaccines. In contrast, antigens recognized by CD8+ T cells are not restricted in their locations, and conserved intracellular molecules could be recognized after antigen presentation. Therefore, the development of malaria vaccines that activate protective CD8+ T cells against blood-stage malaria might be useful and have wide applications.

## Materials and methods

### Mice

C57BL/6 (B6) mice, C57BL/6JSlc-*gld* (*gld*: generalized lymphoproliferative disease) mice and BALB/c mice were obtained from SLC (Hamamatsu, Japan) or Kyudo (Tosu, Japan). *Rag2*−/− mice were obtained from Central Laboratory of Experimental Animals (Kawasaki, Japan). All mice were maintained under specific-pathogen-free conditions. Experiments were generally performed in mice aged 8–12 weeks.

### Ethics statement

All mouse experiments were approved by the Committee for Ethics on Animal Experiments in the Faculty of Medicine, and performed under the control of the Guidelines for Animal Experiments in the Faculty of Medicine, Gunma University and Kyushu University, according to Japanese law (no. 105) and notification (no. 6) of the Government of Japan. No human samples were used in these experiments.

### *Plasmodium yoelii* infection

The clonal lines of blood-stage *P. yoelii* 17XNL (PyNL) and 17XL (PyL) parasites originated from Middlesex Hospital Medical School, University of London, 1984, are generous gifts from Dr M Torii (Ehime University), and the generation of PyNL–GFP has been described previously (*Imai et al., 2013*). Blood-stage parasites were obtained after the fresh passage of frozen stock through a donor mouse, 4–5 days after inoculation. Mice were infected intraperitoneally with 25,000 parasitized red blood cells (pRBCs), unless otherwise indicated.

## Antibodies and reagents

All antibodies were purchased from BD Pharmingen (Franklin Lakes, NJ, USA), eBioscience (San Diego, CA, USA), or BioLegend (San Diego, CA, USA). Fluorescein isothiocyanate (FITC)- and allophycocyanin (APC)-conjugated anti-CD3, FITC-, phycoerythrin (PE)–Cy7-, and APC-conjugated anti-CD8α or β, FITC-, PE–Cy7-, and APC-conjugated anti-CD4, FITC-, and PE- conjugated anti-CD62L, FITC- and PE-conjugated anti-CD44, PE-conjugated anti-CD25, PE-conjugated anti-CD69, PE-, and biotin-conjugated anti-FasL, PE-conjugated anti-LAMP1, PE-, PE-Cy7, biotin-conjugated anti-Fas, FITC-, PE-, PE-Cy7-, and APC-conjugated anti-TER119, PE-, PE-Cy7-, and APC-conjugated anti-MHC class I, PE- and PE-Cy7-conjugated anti-MHC class II, PE- and PE-Cy7-conjugated anti-CD11b, PE- and PE-Cy7-conjugated anti-F4/80, PE- and PE-Cy7-conjugated anti-CD11c, PE- and PE-Cy7-conjugated anti-CD11b, PE- and PE-Cy7-conjugated anti-CD169, PE- and PE-Cy7-conjugated anti-CD68, PE- and PE-Cy7-conjugated anti-SIGNR1, PE-conjugated anti-PanNK, and PE-conjugated anti-Tim4 antibodies were used for flow cytometry. Purified anti-CD16/32 antibodies were used for blocking. Propidium iodide (Sigma, St. Louis, MO, USA) or 7-amino-actinomycin D (BioLegend) were used for dead cell staining, when in some experiments, dead cells were excluded from the analysis. Annexin V (BD Pharmingen) was used to stain PS. Anti-PE-, anti-FITC-, anti-APC-, anti-CD8-, and anti-TER119 microbeads and CD8α$^+$ T cell isolation kit (Miltenyi Biotech, Bergisch Gladbach, Germany) were used for MACS cell purification (Miltenyi Biotech). The PKH26 Red Fluorescent Cell Linker Kit for General Cell Membrane Labeling was from Sigma–Aldrich. The culture medium was RPMI 1640 (Sigma) containing 10% fetal bovine serum, 2 mM L-glutamine, 1 mM sodium pyruvate, 0.1 mM nonessential amino acids, penicillin–streptomycin, and 2-mercaptoethanol. To induce FasL-dependent apoptosis, FasL–*Strep* and *Strep*-Tactin microtiter plates were purchased form IBA (St. Louis, MO, USA).

## Flow cytometry and fluorescence microscopy

For the analysis of white blood cells (WBCs), but not erythroid cells, and in vivo phagocytosis, spleen cells were added to ACK lysing buffer (8024 mg/l NH$_4$Cl, 1001 mg/l KHCO$_3$, 3.722 mg/l EDTA Na$_2$·2H$_2$O) to remove the RBCs. Cell suspensions were incubated with anti-CD16/32 antibody (Fc block) and stained with fluorochrome-labeled antibodies. Isotype control antibodies were also used to evaluate specific staining. Propidium iodide (Sigma) or 7-amino-actinomycin D (BioLegend) were used to stain dead cells, because dead cells were excluded from the analysis in some experiments. Cells were analyzed with a FACSCalibur or FACSAria II flow cytometer (Becton Dickinson, San Jose, CA, USA), and the data were analyzed with the FlowJo software (Treestar, Ashland, OR, USA). Samples were analyzed using a Biorevo BZ-9000 microscope (Keyence, Osaka, Japan). The data were analyzed with the BZ-II software (Keyence).

## Cell separation

**Splenic erythroid cells:** spleen were perfused with medium, and then single-cell suspensions were incubated with anti-CD16/32 antibody (Fc block), washed, and stained with anti-TER119 microbeads or a combination of APC-conjugated anti-TER119 and anti-APC microbeads. The stained cells were collected with the MACS cell separation system (Miltenyi Biotech). The purity of the separated TER119$^+$ cells was usually >90–95%. **CD8$^+$ T cells:** RBCs were removed from the spleen with ACK. The cells were Fc-blocked, then negatively sorted with CD8α$^+$ T cell isolation kit (CD4, CD11b, CD11c, CD19, CD45R (B220), CD49b (DX5), CD105, Anti-MHC-class II, TER119, TCRγ/δ), followed by positive sorting with anti-CD8 microbeads. The purity of the separated CD8$^+$ T cells was usually around 95%. RBCs: Peripheral blood samples were added to a CF11 cellulose column (Whatman, Kent, UK) for the depletion of WBCs and allowed to flow through under gravity. Malaria-parasitized RBCs (pRBCs) were then separated with Percoll density gradient centrifugation (GE Healthcare Bio-Sciences, Piscataway, NJ, USA). In some experiments, the anti-TER119 MACS cell separations system was used to purify the peripheral blood RBCs.

## In vivo depletion of T-cell subsets, prime–boost vaccination, and cell transfer experiments

The depletion of CD4$^+$ or CD8$^+$ T cells was performed as previously described (*Imai et al., 2008*). Briefly, mice were intraperitoneally administered 0.5 mg of anti-CD4 (clone: GK1.5), anti- CD8α (2.43),

or anti- CD8β (YTS156.7.7) antibodies 1 day before and 14 and 28 days after PyNL infection. The depletion of each T-cell subset was checked by flow cytometry, and >99% of the appropriate cell subset was depleted in the peripheral blood by 24 hr after inoculation (*Figure 1—figure supplement 1A*). The depletion of splenic CD8+ T cells in malaria-infected mice is shown in *Figure 1—figure supplement 1B*. The protocols for the prime–boost live vaccination and cell transfer are shown in *Figure 1D*. CD8+ T cells were isolated from WT and *gld* mice infected with PyNL (25,000 pRBC) after two boosts with PyL (50,000 pRBC) at 6 and 9 weeks after the primary PyNL infection. Then, $1 \times 10^7$ purified cells were transferred to recipient x-ray-irradiated (5.5 Gy) mice or *Rag2$^{-/-}$* mice. The recipients were infected with PyL (50,000 pRBC) 1 week after cell transfer.

### In vitro phagocytosis assay

The collected RBCs or pRBCs were washed twice with medium. The cells ($2 \times 10^7$ cells/ml) were stained with 250 nM carboxyfluorescein succinimidyl ester (CFSE: Molecular Probes; Life Technologies, Carlsbad, CA, USA) for 1 min. Staining was stopped by the addition of fetal calf serum, and the cells were washed three times with medium. Splenic CD11b+ macrophages from uninfected mice were sorted with the MACS cell separation system, and the labeled with PKH26, according to the manufacturer's instructions. Splenic CD11b+ macrophages ($1 \times 10^5$ cells) were cocultured with CFSE-labeled pRBCs or normal RBCs in a 1:30 ratio, at a final volume of 200 μl for 4 hr at 37°C in a $CO_2$ incubator with culture medium. Following coculture, the noningested RBCs were removed with ACK lysing buffer. The capacity of macrophages to phagocytize CFSE-labeled pRBCs or normal RBCs was analyzed with a FACSCalibur flow cytometer. For the in vitro phagocytosis inhibition assay, anti-Tim-4 antibody and its isotype control antibody were added to the test sample.

### In vitro PS externalization test

Sorted erythroid cells ($3 \times 10^5$ cells/well) from *gld* mice 17 days after infection with PyNL–GFP were cocultured with CD8+ T cells from WT or *gld* mice 17 days after PyNL infection or from uninfected WT mice at 37°C for 4 hr in a $CO_2$ incubator with culture medium. Effector (CD8): The target (erythroid) ratio was 0:1–10:1. The cells were Fc-blocked and stained with PE-Cy7-conjugated anti-TER119 antibody. PS was then stained with PE-conjugated annexin V in calcium-containing annexin V binding buffer (BD Pharmingen). The parasitized cells (TER119+ GFP+) or unparasitized cells (TER119+ GFP−) were analyzed for PS expression with flow cytometry.

### In vitro PS externalization with FasL–*Strep*

FasL–*Strep* (0–100 ng/ml) was added to a *Strep*-Tactin microtiter plate, and incubated for 20 min at 37°C. Sorted erythroid cells from PyNL–GFP-infected *gld* mice ($2 \times 10^5$ cells/well) were cultured for 4 hr at 37°C in the abovementioned plate. The PS was then surface stained with PE-conjugated annexin V in annexin V binding buffer. In some cases, cells were Fc-blocked and stained with APC- or PE-Cy7-conjugated anti-TER119 antibody and PE-Cy7-conjugated anti-MHC class I antibody, and then analyzed with flow cytometry.

### In vivo depletion of macrophages

Macrophage depletion methods have been previously described (*Van Rooijen and Sanders, 1994*; *Ishida et al., 2013*). Mice were intravenously injected with clodronate (Sigma) liposome (C/L: 1.5 mg clodronate/300-μl liposome suspension) 3 and 9 days after PyNL infection.

### Statistical analysis

Two sets of data (control vs experimental group) were compared and Mann–Whitney U-test was used for statistical analysis. A p-value of $p < 0.05$ was considered to be statistically significant. Significant differences in survival were tested with a log-rank test using Kaplan–Meier survival curves.

## Acknowledgements

This study was supported by Grants-in-Aid (24117504 to HH, 24790399, 26860276 to TI) and the Strategic Fund for the Promotion of Science and Technology to HH from the Ministry of Education, Culture, Sports, Science and Technology of Japan; the Ministry of Health, Labour and Welfare of Japan (H24-Shitei-004) to HH; and the Takeda Memorial Foundation to HH.

## Additional information

### Funding

| Funder | Grant reference | Author |
|---|---|---|
| Japan Society for the Promotion of Science (JSPS) | Grants in Aid 24117504 | Hajime Hisaeda |
| Japan Society for the Promotion of Science (JSPS) | Grants in Aid 24790399 | Takashi Imai |
| Japan Society for the Promotion of Science (JSPS) | Grants in Aid 26860276 | Takashi Imai |
| Ministry of Education, Culture, Sports, Science, and Technology (MEXT) | Promotion of Science and Technology | Hajime Hisaeda |
| Ministry of Health, Labour and Welfare | H24-Shitei-004 | Hajime Hisaeda |
| Takeda Science Foundation | Takeda Memorial Foundation | Hajime Hisaeda |

The funders had no role in study design, data collection and interpretation, or the decision to submit the work for publication.

### Author contributions

TI, Planned and performed the experiments, Analyzed the results, Created the figures, Wrote the manuscript; HI, Generated the recombinant parasites; KS, Analyzed the results; TT, HO, CS, Performed the experiments; HH, Directed the project, Analyzed the results, Wrote the manuscript

### Ethics

Animal experimentation: All mouse experiments were approved by the Committee for Ethics on Animal Experiments in the Faculty of Medicine, and performed under the control of the Guidelines for Animal Experiments in the Faculty of Medicine, Gunma University and Kyushu University, according to Japanese law (no. 105) and notification (no. 6) of the Government of Japan. No human samples were used in these experiments.

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
