## [Decision Letter]

Thank you for sending your work entitled “Cytotoxic activities of CD8^+^ T cells collaborate with macrophages to protect against blood-stage murine malaria” for consideration at *eLife*. Your article has been favorably evaluated by Tadatsugu Taniguchi (Senior editor), a Reviewing editor, and two reviewers.

The Reviewing editor and the reviewers discussed their comments before we reached this decision, and the Reviewing editor has assembled the following comments to help you prepare a revised submission.

Major issues:

1) The statistical analysis is inadequate. The authors did not demonstrate that their data follow a normal distribution. Unless they do so, they should use nonparametric tests such as Mann-Whitney test (if two sets of data are shown and compared; Figures 1C and 1D, 2A, 2B, 2D and 2F, 4B and 4C, 6B, 10D and 10E, 12C and 12D, 13C) or Kruskal-Wallis test (if three sets of data are shown and compared; Figures 5C and 5E, 7C, 9B and 9C, 11A and 11B, 13A).

2) The studies described in Figures 1 and 2 showing the effect of T cell depletion in mice (CD8 or CD4 depleted) are critical as they provide information about how important are CD8^+^ T cells for protection and parasite clearance. Unfortunately they are poorly described: no information on depletion schedule or the amount of antibodies. The results of Figure 1 indicate that most CD8^+^ T cell-depleted animals control parasitemia and clear parasites, suggesting only a partial, modest role for these cells. In Figure 2, a similar experiment is shown that suggests a more critical role for CD8^+^ T cells, as only 20 % of the depleted-mice survive. It is not clear how this experiment differs from those shown in Figure 1. The strain of mice is not stated, and the antibody used, the depletion schedule and the amounts are not described. Please clarify how the experiments were performed and address the substantial differences in outcome for what are superficially the same types of experiments.

3) The results in Figure 2 show that adoptive transfer of purified immune 10 7 CD8^+^ T cells into irradiated mice confers protection to parasite challenge. This conclusion would be valid if the transferred cells contain only CD8^+^ T cells. The authors purify cells by magnetic sorting, which does not provide “pure” cell populations; it is likely that such preparations contain cells from different sub-sets. In irradiated animals, a very few contaminant T cells can proliferate intensively and acquire effector capacity. It would be important to demonstrate that 5 and 10 days after challenge, mice that received purified CD8^+^ T cells do not have expansion of other cell subsets.

4) If phagocytosis is an important mechanism as suggested by the authors, it would be informative to determine what happens if phagocytes are depleted a few days after infection. Likewise, controls for the effects of the anti-TIM1 antibody on uptake are needed, especially given the modest effect seen in Figure 13. Finally, some immunohistochemical staining analysis to show uptake of parasites by macrophages and the subsets of macrophages involved in the spleen in animals with and without CD8 T cell effectors would be especially valuable in supporting and extending the major claims of the paper.

5) The authors claim in the Results associated with Figure 2 that signs of CD8 T cell activation indicate a role for these cells in host protection. This is not correct. Activation could occur without the resulting cells being able to provide useful host defense. Drawing this conclusion requires the depletion and transfer experiments shown later, and should not be given as an interpretation at this point in the paper.

6) No mention is made of how macrophages become activated to be parasiticidal. Is IFNg involved and if so, does it come from the activated CD8^+^ T cells? Also, the data do not really show direct activation of the CD8 T cells by infected RBC in vivo; this could occur through cross-presentation of the blood stage antigens by phagocytes that have taken up infected cells and interacted with the CD8 T effectors; the authors would need to use bone marrow chimeras with two distinct MHC types and transfer of effector CD8 T cells from one or the other donors to test whether there is selective destruction of infected RBC of only the correct MHC type to show a direct interaction and contact dependent Fas-Fas mediated induction of PS expression to properly support the model they propose.

---

## [Author Response]

*1) The statistical analysis is inadequate. The authors did not demonstrate that their data follow a normal distribution. Unless they do so, they should use nonparametric tests such as Mann-Whitney test (if two sets of data are shown and compared; Figures 1C and 1D, 2A, 2B, 2D and 2F, 4B and 4C, 6B, 10D and 10E, 12C and 12D, 13C) or Kruskal-Wallis test (if three sets of data are shown and compared; Figures 5C and 5E, 7C, 9B and 9C, 11A and 11B, 13A)*.

As the reviewers suggested, we have reanalyzed statistical differences between two sets of data using the Mann-Whitey U-test, as described both in the Materials and methods and the figure legends.

*2) The studies described in*
Figures 1 and 2
*showing the effect of T cell depletion in mice (CD8 or CD4 depleted) are critical as they provide information about how important are CD8*^*+*^
*T cells for protection and parasite clearance. Unfortunately they are poorly described: no information on depletion schedule or the amount of antibodies. The results of*
Figure 1
*indicate that most CD8*^*+*^
*T cell-depleted animals control parasitemia and clear parasites, suggesting only a partial, modest role for these cells. In*
Figure 2*, a similar experiment is shown that suggests a more critical role for CD8*^*+*^
*T cells, as only 20 % of the depleted-mice survive. It is not clear how this experiment differs from those shown in*
Figure 1*. The strain of mice is not stated, and the antibody used, the depletion schedule and the amounts are not described. Please clarify how the experiments were performed and address the substantial differences in outcome for what are superficially the same types of experiments*.

We have added the cell depletion protocols and information on the mouse strain used in these experiments to the Materials and methods section and to the figure legends.

The experiments in the original Figure 2 (Figure 1 in the revised manuscript) used B6 mice and were exactly the same as in the original Figure 1 (new Figure 1—figure supplement 1), yet the results do differ slightly. One possible reason is environmental influences, as these two experiments were conducted in different animal facilities using mice from different suppliers. In addition, we frequently experienced experimental variations in such infections using mice even under the same conditions (e.g., two or three out of five mice died, giving 40% or 60% mortality, respectively), indicating the limitation of biological experiments. However, it is important that mice depleted of CD8^+^ T cells unfailingly became more susceptible to PyNL infection. To avoid any confusion, we changed the original Figure 1 to Figure 1—figure supplement 1 and Figure 1—figure supplement 2, and described some of the above in the figure legend.

*3) The results in*
Figure 2
*show that adoptive transfer of purified immune 10 7 CD8*^*+*^
*T cells into irradiated mice confers protection to parasite challenge. This conclusion would be valid if the transferred cells contain only CD8*^*+*^
*T cells. The authors purify cells by magnetic sorting, which does not provide “pure” cell populations; it is likely that such preparations contain cells from different sub-sets. In irradiated animals, a very few contaminant T cells can proliferate intensively and acquire effector capacity. It would be important to demonstrate that 5 and 10 days after challenge, mice that received purified CD8*^*+*^
*T cells do not have expansion of other cell subsets*.

As the reviewers suggested, we evaluated the untoward effects of contaminants in transferred CD8^+^ T cells. Furthermore, the dose of CD4^+^ T cells required for protection was determined using *Rag2KO* mice instead of irradiated mice. We used *Rag2KO* recipients because irradiation is hard to do in our current facility. In relation to the second part of this comment, irradiation was done in another facility to which we used to belong.

We added the results as Figure 1—figure supplement 2 and a description to the text.

*4) If phagocytosis is an important mechanism as suggested by the authors, it would be informative to determine what happens if phagocytes are depleted a few days after infection. Likewise, controls for the effects of the anti-TIM1 antibody on uptake are needed, especially given the modest effect seen in Figure 13. Finally, some immunohistochemical staining analysis to show uptake of parasites by macrophages and the subsets of macrophages involved in the spleen in animals with and without CD8 T cell effectors would be especially valuable in supporting and extending the major claims of the paper*.

Following the reviewers’ suggestion, we performed macrophage depletion. It confirmed the importance of phagocytes (see Figure 7).

In the new Figure 10, isotype IgG was used as a control for the anti-Tim-4 antibody. It did not affect the phagocytic activity.

We also performed immunohistochemical analyses using macrophage markers to clearly demonstrate the uptake of parasites by macrophages (Figure 8).

Finally, we analyzed macrophage subsets and found that F4/80^+^ red pulp macrophages are responsible for the ingestion of parasites (Figure 8).

These results were included in the revised version of the manuscript.

*5) The authors claim in the Results associated with*
Figure 2
*that signs of CD8 T cell activation indicate a role for these cells in host protection. This is not correct. Activation could occur without the resulting cells being able to provide useful host defense. Drawing this conclusion requires the depletion and transfer experiments shown later, and should not be given as an interpretation at this point in the paper*.

We agree with the reviewers’ comment, and we have deleted the following sentence, “activation indicates a role for these cells in host protection”, in the Results section.

*6) No mention is made of how macrophages become activated to be parasiticidal. Is IFNg involved and if so, does it come from the activated CD8*^*+*^
*T cells? Also, the data do not really show direct activation of the CD8 T cells by infected RBC in vivo; this could occur through cross-presentation of the blood stage antigens by phagocytes that have taken up infected cells and interacted with the CD8 T effectors; the authors would need to use bone marrow chimeras with two distinct MHC types and transfer of effector CD8 T cells from one or the other donors to test whether there is selective destruction of infected RBC of only the correct MHC type to show a direct interaction and contact dependent Fas-Fas mediated induction of PS expression to properly support the model they propose*.

As the reviewers hypothesized, activation of macrophages may depend on IFN-γ. We already demonstrated that the protective ability of transferred “immune” CD8 T cells depends on IFN-γ responsible for macrophage activation (see Imai et al, Eur. J. Immunol., 2010), which is also described in the Introduction section in the original version of the manuscript. In the present study, however, the remaining CD4^+^ T cells may also produce IFN-γ after depletion of CD8^+^ T cells. In fact, macrophages were activated in response to infection with PyNL even in CD8^+^ T cell-depleted animals as indicated by higher expression of MHC class I and class II (Figure 9). These results support our proposal that the impaired phagocytosis in CD8^+^ T cell-depleted mice is not due to insufficient activation of macrophages but due to the reduction of PS^+^ infected erythroid cells.

It is difficult to analyze whether effector CD8^+^ T cells affect infected erythroid cells directly after recognizing them or indirectly after recognition of dendritic cells cross-presenting parasite antigens in vivo. As both erythroid cells and dendritic cells are derived from bone marrow, bone marrow chimeras would not solve these questions. To compensate for the lack of in vivo evidence, in vitro experiments have been performed and revealed that infected TER119^+^ cells from the spleen, but not peripheral blood, externalized PS when co-cultured with effector CD8^+^ T cells (Figure 4, Figure 5). We also reported previously that effector CD8^+^ T cells are directly activated by infected erythroid cells in an antigen-specific manner using OT-I CD8^+^ T cells that recognize OVA peptide in conjunction with H-2K^b^ and OVA-expressing PyNL parasites (see Imai et al, Sci. Rep., 2013). We therefore believe that direct antigen-specific activation operates, at least partially, during CD8^+^-T-cell-mediated protection.